# Dietary Conjugated Linoleic Acid Reduces Body Weight and Fat in *Snord116^m+/p−^* and *Snord116^m−/p−^* Mouse Models of Prader–Willi Syndrome

**DOI:** 10.3390/nu14040860

**Published:** 2022-02-18

**Authors:** Brittney Knott, Matthew A. Kocher, Henry A. Paz, Shelby E. Hamm, William Fink, Jordan Mason, Robert W. Grange, Umesh D. Wankhade, Deborah J. Good

**Affiliations:** 1Department of Human Nutrition, Foods and Exercise, Virginia Tech, Blacksburg, VA 24060, USA; brittknott25249@gmail.com (B.K.); hshelby@vt.edu (S.E.H.); willy.fink22@gmail.com (W.F.); jmason20@vt.edu (J.M.); rgrange@vt.edu (R.W.G.); 2Translational Biology, Medicine and Health Graduate Program, Virginia Tech, Roanoke, VA 24016, USA; mak428@vt.edu; 3Department of Pediatrics, College of Medicine, University of Arkansas Medical Center, Little Rock, AR 72202, USA; hapazmanzano@uams.edu (H.A.P.); uwankhade@uams.edu (U.D.W.); 4Arkansas Children Nutrition Center, Arkansas Children’s Research Institute, Little Rock, AR 72202, USA; 5Metabolism Core Facility, Virginia Tech, Blacksburg, VA 24060, USA

**Keywords:** *Snord116*, dietary intervention, obesity, muscle function, exercise, microbiome, RNA-seq

## Abstract

Prader–Willi Syndrome (PWS) is a human genetic condition that affects up to 1 in 10,000 live births. Affected infants present with hypotonia and developmental delay. Hyperphagia and increasing body weight follow unless drastic calorie restriction is initiated. Recently, our laboratory showed that one of the genes in the deleted locus causative for PWS, *Snord116*, maintains increased expression of hypothalamic *Nhlh2*, a basic helix–loop–helix transcription factor. We have previously also shown that obese mice with a deletion of *Nhlh2* respond to a conjugated linoleic acid (CLA) diet with weight and fat loss. In this study, we investigated whether mice with a paternal deletion of *Snord116* (*Snord116^m+/p−^*) would respond similarly. We found that while *Snord116^m+/p−^* mice and mice with a deletion of both *Snord116* alleles were not significantly obese on a high-fat diet, they did lose body weight and fat on a high-fat/CLA diet, suggesting that the genotype did not interfere with CLA actions. There were no changes in food intake or metabolic rate, and only moderate differences in exercise performance. RNA-seq and microbiome analyses identified hypothalamic mRNAs, and differentially populated gut bacteria, that support future mechanistic analyses. CLA may be useful as a food additive to reduce obesity in humans with PWS.

## 1. Introduction

Prader–Willi Syndrome (PWS) is a genetic condition that occurs in up to 1 in 10,000 live births [1]. Individuals with PWS show initial developmental delay, significant hypotonia/muscle weakness, and typically demonstrate some level of intellectual disability, and obesity later in childhood or adolescence. The most common cause of the syndrome is a de novo deletion of the paternal 15q chromosome, as the maternal allele is imprinted and not expressed. Uniparental (maternal) disomy and an imprinting locus mutation can also be causative. A minimal deletion of chromosome 15q that causes PWS includes just two expressed regions: the *SNORD116* cluster (a group of 28 or more small nucleolar RNAs “snoRNAs”), and the *IPW* gene which also encodes a non-coding RNA of little known function [2]. For PWS patients, there is, as yet, no cure, and few treatment options.

In late 2016, Burnett and colleagues showed that *NHLH2* and *PCSK1* (PC1/3) mRNAs and proteins were downregulated in PWS-derived induced pluripotent stem cell neurons (iPSC), and in the PWS mouse model, the *Snord116**^m+/p^*^−^ mouse [3]. Likewise, PWS patients have a 1.5-fold reduction in *NHLH2* expression in lymphoblastoid cells [4]. We recently demonstrated that *Nhlh2* mRNA is indeed upregulated post-transcriptionally by *Snord116* snoRNA [5].

Interestingly, mice with a deletion of *Nhlh2* (N2KO mice), which we developed in 1997 [6], share multiple phenotypes with the PWS *Snord116**^m+/p^*^−^ mouse model, which has a deletion of *Snord116* only on the paternally inherited allele [7]. However, the *Snord116**^m+/p^*^−^ mouse (PWS mouse) fails to develop overt obesity on regular mouse chow, but they may develop hyperphagia by three months of age [7]. A mouse model with hypothalamic-only adult deletion of *Snord116* does develop both hyperphagia and later-onset obesity [8], and there are ~25 different PWS mouse models containing genetic manipulation with the mouse chromosome 7 regions homologous to human 15q—each with varying similarity to the human condition [9]. The PWS *Snord116**^m+/p^*^−^ mouse model was chosen as a model for this study as it is most similar genetically to the minimal deletion in humans that results in PWS phenotypes, and a mouse with hypothalamic adult *Snord116* conditional deletion develops increased adiposity and hyperphagia [8]. Additionally, the PWS *Snord116^m+/p^*^−^ mouse model has most recently been used in a study to test the use of growth hormone therapy as a treatment option for PWS patients [10]. Finally, Qi and colleagues previously published on the body weight and food intake characteristics of the PWS KO mouse (*Snord116^m^*^−/*p*−^) [11], which we included in this study to analyze complete absence of the *Snord116* alleles. These results could mimic an individual carrying one deleted *Snord116* allele, along with an inactivating variant in *Snord116* or another PWS locus gene. One such patient with a homozygous variant in SNURF-SNRPN gene in the PWS locus has been described [12].

Previously, we demonstrated that conjugated linoleic acid (CLA) treatment of N2KO mice on a 20% fat diet led to weight loss, decreased body fat, increased metabolism, muscle mitochondrial biogenesis, and increased exercise performance [13,14,15,16]. At the time of the initial paper, these outcomes represented the first time that genetic obesity was “cured” with CLA. CLA is currently considered to be safe (GRAS, “generally regarded as safe”) by the FDA and is available over the counter. However, results from human studies have been variable, which could be due to differences in dosage, human genetics, and small study sample sizes [17].

Aside from growth hormone [18], there are few treatment options for individuals with PWS. Given the overlap in phenotypes between the N2KO, *Snord116**^m+/p−^* mouse model, and PWS in humans, and the finding that *Nhlh2* is downstream of *Snord116*, we sought to determine if deletion of *Snord116* in the *Snord116**^m+/p−^* mouse model affected the response to dietary CLA. With later-onset obesity and inactivity being clinically relevant features of PWS, CLA could prove to be a potential cost-effective treatment for patients.

## 2. Materials and Methods

### 2.1. Mouse Housing, Breeding, and Euthanasia

All animal protocols were approved by the Institutional Animal Care and Use Committee at Virginia Tech. Mice with the *Snord116* paternal deletion (B6[Cg]-Snord116tm1.1Uta/J Stock No: 008149|1-loxp (KO), Snord116del) were obtained from Jackson Laboratories and maintained on a C57Bl/6 background. Genotyping and breeding were performed as reported [7]. WT mice for the study were siblings of either the PWS (*Snord116^m+/p^**^−^*) or PWS-KO (*Snord116^m^**^−^**^/p−^*) mice. PWS mice were generated by crossing WT females and male mice who were *Snord116^m^**^−^**^/p+^*. Breeder males (*Snord116^m^**^−^**^/p+^*) were generated in separate crosses to inherited their deleted allele from a maternal chromosome so that they were considered phenotypically normal in regards to *Snord116* expression. PWS-KO (*Snord116^m^**^−^**^/p−^*) mice were generated by using non-sibling male and female mice of the genotype *Snord116^m^**^−^**^/p+^* who had inherited their deleted allele maternally. In this type of mating, 50% of the offspring would be heterozygous (and never used for mating or the study, as the origin of their deleted allele could not be determined), and 25% each would have WT alleles or would be PWS-KO. Mice were housed at room temperature at about 22 °C with 12 h light/dark cycles at 7 a.m. and 7 p.m. and ad-libitum access to food (4.5% crude fat) and water. Mice were weaned at 3 weeks old, genotyped using an ear snip, ear tagged, and housed with littermates of the same sex. For tissue dissection, mice were euthanized by CO_2_ asphyxiation between 12 p.m. and 2 p.m.

### 2.2. Study Design

At 8 weeks of age, male mice were randomly assigned to treatment groups and separated into individual cages (Table 1). As shown in Figure 1, this timepoint was designated as pre-week and baseline measurements were recorded. At Week 1 of the study, a diet containing 20% fat alone, or with conjugated linoleic acid (CLA), was provided. CLA was obtained as a gift from BASF Chemical Company, Florham Park, NJ, USA. Diets were formulated by Envigo Teklad (Madison, WI, USA) under the manufacturing numbers TD.190458 Rx 2984761 and TD.190459 Rx 2984779. The control diet contained 950 g of TD.05350 with 50 g of soybean oil and the CLA diet contained 950 g of TD.05350 with 45 g of soybean oil and a 5 g 50:50 mixture of the trans-10,cis-12 isomer and trans9, cis-11 isomer of CLA supplied by BASF Chemical Company (trade name Tonalin^tm^, 0.5% of food composition). This formulation matches that of previous studies conducted with the *Nhlh2* knockout mice, e.g., [13], providing approximately 0.9 g of CLA per week based on the average mouse food intake in the study, and 0.13 g of CLA weekly. Calculated to an average mouse body weight of 25 g, this amount is equivalent to 5.2 g per kg body weight daily.

### 2.3. Weekly Measurements

Body weight, fat, and lean mass, rectal temperature, and food intake were measured weekly. For body weight, fat, and lean mass measurements, mice were weighed on Mondays using a standard scale, and then placed in a Bruker (Billrica, MA, USA) LF90 NMR machine using the mouse holder. Measurements were recorded in grams for fat and lean mass. Temperature measurements were performed in triplicate on Mondays using a YSI pediatric rectal probe attached to a TH-5 Thermalert Monitor (Physitemp Instruments, Clifton, NJ, USA). Food intake measurements were made twice weekly (Mondays and Thursdays) and total weekly intake was calculated and reported alone, as well as per gram body weight.

### 2.4. Study Pre- and Post-Measurements

Wheel running, elevated plus maze anxiety-like behavior testing, and glucose tolerance measurements were conducted in both the pre- and post-weeks of the study. Each are described separately below.

#### 2.4.1. Wheel Running

For the spontaneous/motivated wheel running analysis, mice were placed in wire-bottom cages equipped with computer-monitored running wheels (Mini-mitter, Sunriver, OR, USA) for a total of 4 days, and then returned to their home cages. The first two days were not recorded as these were considered acclimation days, and the last 48 h were used for data collection. Mice had ad lib access to food and water during this time. For the pre-week, regular mouse chow was provided. For the post-week measurements, mice were supplied with their study diet during testing.

#### 2.4.2. Elevated Plus

A plus-shaped elevated plus apparatus with two open and two closed arms was constructed using measurements obtained from a commercial apparatus. This elevated plus maze was used for anxiety-like behavior analysis [19]. The maze was cleaned with 70% ethanol before and after the procedure for each mouse and allowed to fully dry between mice. Mice were placed in the center/cross-arms and allowed to freely explore the maze for 5 min while the duration and frequency of entries into open and closed arms was recorded. Animals that enter the open arms more frequently are considered to display less overall anxiety-like behavior than those that stay in the closed arms [19].

#### 2.4.3. Glucose Tolerance Tests

Mice were food-deprived for 12–15 h during the dark cycle in a cage devoid of bedding material. For the fasting measurement, the tail was snipped (1–2 mm), and blood was collected directly on Care Touch glucose test strips and directly measured using a Care Touch blood glucose monitor (Future Diagnostics USA, Brooklyn, NY, USA). Glucose (pharmaceutical grade dextrose, 2 g/kg in PBS, sterile) was injected intraperitoneally (IP) and tail blood samples were obtained at 15, 30, 60, 90, and 120 min following the injection. Area under the curve was calculated using the formula (((([fasting glucose × 7.5] + [15 min glucose × 15) + [30 min glucose × 22.5]) + [60 min glucose × 30]) + [90 min glucose × 30]) + [120 min glucose × 15]).

### 2.5. End-of-Study Measurements

Rotarod balance measurements, metabolic measures, mouse functional muscle testing, euthanasia with blood collection, tissue collection, and histology were all performed only as post-measurements at the end of the study during Weeks 13–14. Mice continued on their study diet until euthanasia.

#### 2.5.1. Rotarod Analysis

For rotarod analysis, animals were tested using a Columbus Instruments (Columbus, OH, USA) Economex Rota-Rod apparatus. All mice in the study were given four trials each day during the day (between 11 a.m. and 1 p.m.) for four consecutive days from 0 rpm to a maximum speed of 20 rpm, with an acceleration slope of 2.65%. Animals were tested for a maximum of 5 min of running per test, or until the animals fell off the device. The Economex Rota-Rod apparatus is equipped with a pressure-sensitive landing area so that the time spent on the rotating rod is automatically recorded when the animal falls. The procedure has been previously described by our laboratory [20]. Animals were given between 10 and 15 min of rest in between trials. Note that there are four days of testing, with each day consisting of four trials. The initial trials on Day 1 use a lane where we placed non-slip bathroom tape to provide a stable environment for the animal to learn the procedure. The non-taped lanes were used for Days 2–4. Days 2–3 serve as acclimation days. Day 4 data were used for analysis.

#### 2.5.2. Indirect Calorimetry and Home-Cage Activity

Indirect calorimetry and locomotor activity measurements were performed using a Labmaster Mouse Calorimetry and Locomotor system (TSE Systems, Bad Homburg, Germany) in the Metabolism Core Virginia Tech. VO_2_ consumption and VCO_2_ production in individual mice were measured using metabolic chambers. Air going into the TSE system was at 20.9% oxygen, 0.05% CO_2_, and the airflow rate was 0.4 L/min (Airgas, Christiansburg, VA, USA). Data were collected every 15 min. Body composition was measured, as described above, prior to assessment of the animals using the TSE system. A photobeam-based activity monitoring system detected and recorded ambulatory movements. The results were used to calculate the respiratory exchange ratio (RER) and total energy expenditure/gram lean mass. Energy expenditure (kJ/h) was calculated using the formula VO_2_ × (3.815 + (1.232 × RER)) × 4.1868 [21] and normalized to the lean mass determined by NMR. All parameters were measured continuously and simultaneously for 48 h after approximately 20 h of adaptation for single-housed mice. The average values for the last 24–48 h were used for analysis. To calculate hourly activity, the average of two-hour bins of activity from the last 24 h period was plotted. Activity level was plotted separately as time of rest (activity level = 0 beam breaks per 15 min bin) or activity (activity level equal to or greater than 100 beam breaks per 15 min bin) and summed for the 24 h period. Due to equipment malfunction occurring during the study, calorimetry was performed on only the following numbers for each genotype and treatment: *N* = 7 WT control, *N* = 5 WT CLA, *N* = 6 PWS control, *N* = 6 PWS CLA, *N* = 4 PWS-KO control, *N* = 4 PWS-KO CLA. These reduced *n* values may have contributed to the non-significant finding for the indirect calorimetry assessment.

#### 2.5.3. In Vivo Muscle Function Testing

The procedures for in vivo muscle testing were recently published [22]. Briefly, at the end of the study, following both running wheel and metabolism measurements, body weight was determined, and mice were anesthetized with isoflurane (VetOne Fluriso, Boise, ID, USA) and placed on the temperature-controlled platform (40 °C) of the contractile apparatus (ASI), as described. The right hindlimb was shaved, hair remover applied (Nair Hair Remover Lotion, Ewing, NJ, USA) for 30 s, cleaned with 2-inch × 2-inch gauze and tap water, and swabbed with povidone-iodine (Betadine Solution Swabsticks, Stamford, CT, USA). The knee was clamped so the tibia was 90° to the femur. The foot at 90° to the tibia was secured with clear Transpore surgical tape (M3, St. Paul, MN, USA) to the foot pedal of the Aurora Scientific (ASI; Aurora, ON, Canada) dual-mode servomotor. The mouse tail was taped (M3, St. Paul, MN, USA) loosely to the platform to keep it clear of the foot pedal. In vivo plantarflexor torque–frequency and fatigue assays were determined as described in Hamm et al. [22]. Dynamic Muscle Control (DMC) software controlled the timing and frequency of the stimulations and collection of torque. Peak torque for each muscle contraction was determined using ASI Dynamic Muscle Analysis (DMA) software. Due to the body mass and fat mass reduction of the CLA groups in this study, torque was normalized separately by body mass (g) and the final measure of lean body mass (g).

### 2.6. RNA Isolation

Fresh hypothalamus tissue was lysed in TRIzol^®^ using a rotor stator homogenizer. TRIzol samples were frozen at −20 °C in microfuge tubes until purified (2 weeks to 5 months). Thawed TRIzol samples were purified using the TRIzol + Purelink RNA minikit (ThermoFisher, Waltham, MA, #12183025), following manufacturer’s instructions for the TRIzol^®^ Plus Total Transcriptome Isolation protocol. Purified RNA was then DNAse-treated using TURBO DNA-free™ Kit (ThermoFisher #AM1907) according to manufacturer’s instructions, diluted to 60 ng/μL in nuclease-free water, and stored at −80 °C.

### 2.7. RNA-Seq

RNA sequencing and library preparation was performed by Virginia Tech’s Genomics Sequencing Center facility at the Fralin Life Sciences Institute. Total RNA with an RNA Intensity Number ≥ 8.0 was converted into a strand-specific library using Illumina’s TruSeq Stranded mRNA HT Sample Prep Kit (Illumina, San Diego, CA, RS-122-2103), for subsequent cluster generation and sequencing on Illumina’s NextSeq. The library was enriched by 14 cycles of PCR, validated using Agilent TapeStation, and quantitated by qPCR. Individually indexed cDNA libraries were pooled and sequenced on NextSeq 75 SR. The Illumina NextSeq Control Software v2.1.0.32 with Real Time Analysis RTA v2.4.11.0 was used to provide the management and execution of the NextSeq 500 and to generate binary base call (BCL) files. The BCL files were converted to FASTQ files, adapters trimmed, and demultiplexed using bcl2fastq Conversion Software v2.20. FASTQ files were aligned to mouse genome GRCm38.p6 using the Geneious RNA assembler 2 January 2020 from Geneious Prime with map quality 30 (99.9% confidence) and spanning intron annotations. Raw read counts were quantified to gene and/or transcript annotations with loss of strand-specificity. Differential expression analysis across experimental conditions was performed using the DESeq2 plugin in Geneious Prime [23,24,25]. Default settings for filtering low-expression mapped genetic features were used for the DESeq2 plugin. False discovery rate (FDR) was calculated using the Benjamini–Hochberg procedure with a threshold for significance of FDR < 0.10. Additionally, raw read counts were used to quantify differential expression through edgeR and voom using the DEApp online webserver [26]. Low-expression mapped genetic features were removed with log2CPM < 1 in 2 or more samples. The results from the three differential expression methodologies, DESeq2, edgeR, and voom, were compared using R and spreadsheet software such as Microsoft Excel. Differentially expressed genes were analyzed for gene-ontology enrichment with MouseMine [27].

### 2.8. Reverse-Transcriptase Quantitative PCR

For RT-QPCR, a Power SYBR^®^ Green RNA-to-CT™ 1-Step Kit (ThermoFisher #4389986) was used according to manufacturer’s instructions. Reactions of 10 μL were performed using 150 nM final primer concentration. Primers were assessed for efficiency using a dilution series and fell within 90–110% efficiency. A 90 ng measure of RNA was used per 10 μL reaction. Two to three technical replicates were performed. Control reactions for each sample (minus reverse-transcriptase and minus template controls) were used for quality control. On the ViiA 7 Real-Time PCR System (ThermoFisher), 384-well plates were run according to RT-QPCR mix instructions, and thermocycling conditions were not modified from suggested protocol (one-step annealing/extension at 60 °C). Quality-control measures including melt-curve analysis, technical replicate analysis, etc., were analyzed by thermocycler software and by operator; any major errors were excluded from analysis when deemed appropriate by the quality-control software, and/or new samples and plates were run. Candidate reference genes for ddCT analysis were analyzed and mouse beta-actin was chosen as the reference gene control for all experiments.

### 2.9. Microbiome Analysis

Cecal samples were flash-frozen in liquid nitrogen and stored at −80 °C. DNA was isolated from cecal content using a DNeasy PowerSoil Pro Kit (Qiagen #47014) and the TissueLyser II (Qiagen #85300). Fifty nanograms of genomic DNA was utilized for amplification of the V4 variable region of the 16S rRNA gene using 515F/806R primers. Forward and reverse primers were barcoded to accommodate multiplexing up to 384 samples per run as described by Kozich and colleagues [28]. Paired-end sequencing (2 × 250 bp) of pooled amplicons was carried out using an Illumina Miseq platform with ~30% PhiX DNA.

Demultiplexing, adapter trimming, and generating the fastq files were performed automatically using the Miseq Reporter on the instrument computer. Bioinformatics analysis was then conducted using the QIIME 2 platform [29]. Denoising was performed with an initial quality filtering followed by the Deblur algorithm [30,31]. Representative amplicon sequence variants (ASVs) were used to generate a phylogenetic tree with FastTree [32], and taxonomy was assigned using a Naive Bayes classifier trained on the Greengenes 13_8 reference [33]. To evaluate diversity metrics, samples were rarefied to an even depth of 4397 quality-filtered reads. Beta diversity was assessed using the weighted Unifrac distances [34] and visualized using the principal coordinate analysis (PCoA) plot.

### 2.10. Histology

Tissues were isolated immediately following euthanasia by CO_2_ asphyxiation and placed into 4% paraformaldehyde, overnight, with rocking at 4 °C. The tissues were then rinsed in 70% ethanol and stored in 70% ethanol at 4 °C until processing. The Virginia Tech Veterinary Teaching Hospital at the Virginia-Maryland College of Veterinary Medicine processed the tissues for histology and stained with hematoxylin–eosin stain for microscopy. Representative samples were visualized using a 40× ocular on a Nikon Eclipse 50i microscope and captured using an Olympus Q-color3 camera.

### 2.11. Statistical Analyses

All values were expressed as mean ± SEM unless indicated otherwise. Comparison of means between groups and calculation of *p*-values were made using JMP Pro15 software (Cary, NC, USA), with Tukey’s post hoc analysis for multiple comparisons when overall *p*-values in the comparison were significant. For the responses of weight, fat, lean mass, temperature, and food intake shown in figures, a fitted least squares regression with effects of genotype, treatment, and genotype × treatment was analyzed for Week 12 data only. Responses of these dependent variables were also analyzed across the 12-week time period, adding in effects of week, week × treatment, and week × genotype. These results are shown in Appendix A as individual data tables. The responses that had pre–post measures (wheel running, elevated plus, fasting glucose, glucose tolerance) were analyzed separately for the pre and post time periods (no effect of time). Animals were grouped together for the pre time period, regardless of their assigned data, as all animals were on standard mouse chow at that time. A fitted least squares regression with effects of genotype, treatment, and genotype × treatment was analyzed for these data, and Tukey post hoc analysis performed when effects were significant. Rotarod data were collected only during the post-period and analyzed with a response of time and effects of genotype, treatment, and genotype × treatment. For area-under-the-curve calculations, the formula function in JMP was used to calculate the area for individual animals pre and post, and then fit least squares regression model used for effects of genotype, treatment, and genotype × treatment.

All RT-QPCR data were analyzed using Microsoft Excel 16 for Microsoft 365, IBM SPSS Statistics 26 for Windows, and GraphPad Prism 9.0.0. The numbers of samples in statistical tests are described in respective figures. The 2^ddCT^ method of relative quantification was used. Statistical significance tests were performed on respective ddCT values, from which Relative Quantification values were derived. A two-way ANOVA with Bonferroni correction was used for relative expression of RNA normalized to WT or control conditions.

In vivo plantarflexor torque–frequency and fatigue curves were analyzed using GraphPad Prism 9.2.0. A two-way ANOVA was performed with multiple comparisons across groups. For torque–frequency figures, a nonlinear model was fit using the sigmoidal curve with variable slope. 

Statistical analyses for microbiome data were performed in R v4.0.5. Differences in beta diversity were evaluated using permutational multivariate analysis of variance (PERMANOVA) with 999 permutations and including the weighted Unifrac distance matrix. The Kruskal–Wallis test was used to evaluate alpha metrics and taxonomic differences and the Dunn’s test was used to evaluate pairwise multiple comparisons, with *p*-values corrected for multiple testing using the false discovery rate (FDR) approach [35]. 

## 3. Results

### 3.1. Body Weight and Fat Mass Are Reduced in PWS and PWS-KO Mice

The 20% fat diet only results in a slight body weight and fat increase over a normal chow diet, and the highest weight gain occurs in the WT animals on control diet (Figure 2A,B). Over the 12 weeks of the study, the effects of genotype (*F* = 234.19, *p* < 0.001), treatment (*F* = 179.63, *p* < 0.001), and week (*F* = 8.75, *p* < 0.001) were all highly significant. Additionally, the cross effect of genotype × treatment was significant (*F* = 4.49, *p* = 0.0117), with post hoc analysis indicating significant differences between WT on control diet and all other genotypes and treatments (Figure 2A). In the effect summary for body weight, response of genotype, treatment, and week were highly significant (*p* < 0.001). Treatment by week response was significant (*p* = 0.044), as was genotype by treatment (*p* = 0.016). Analysis of the end of study (Week 12) data for body weight indicates that there was a significant effect of genotype (*F* = 17.39, *p* < 0.001) and treatment (*F* = 23.4, *p* < 0.001). Post hoc analysis of genotype shows significance (Figure 2B). Although PWS and PWS-KO mice on the HFD weighed significantly less than WT mice (Appendix A), CLA diet still caused significant reductions in body weight overall for the three genotypes (no genotype × treatment effect, but a significant effect of treatment, Appendix A). Post hoc analysis for body weight (Figure 2A) indicated that there were significant body weight reductions for each genotype on the CLA diet, as compared to control diets (*F* = 3.70, *p* = 0.025).

For body fat measures, over the 12 weeks of the study, the effects of genotype (*F* = 109.77, *p* < 0.001), treatment (*F* = 619.05, *p* < 0.001) (Appendix A), week (*F* = 2.15, *p* = 0.0132), and genotype by week (*F* = 32.62, *p* < 0.0001) were all highly significant, while the interaction of genotype by week was not (*p* = 0.8962), indicating that genotypes all responded to the treatment similarly. In analyzing body fat at the end of the study, WT mice on the control diet had ~4 g more body fat than WT mice on the CLA diet. Comparing fat levels to differences in body weight indicate that increased weight in control-diet groups was nearly all due to increased body fat (Figure 2B,C). Post hoc analysis of CLA treatment effects overall (all genotypes) on body weight indicates there was a 3.6-fold reduction in body fat levels with treatment, compared to control HFD alone (Appendix A).

Lean mass measures muscle and organ weight, which consists of muscle, organs, bones, and fluids, constituting most of an animals’ body weight. The overall effect of genotype on lean mass in the whole model was highly significant (*F* = 52.95, *p* < 0.001), without a treatment effect (*F* = 0.003, *p* = 0.96), but with both an effect of week (*F* = 11.30, *p* = <0.001,), and an interaction of genotype × treatment (*F* = 4.81, *p* = 0.0085,). No other interactions were significant. Post hoc analysis of the genotype × treatment interaction through the entire study indicated that CLA-treated PWS-KO and PWS mice had lower overall lean mass when compared to CLA-treated WT mice (Appendix A). Likewise, WT control mice (19.73 g) had increased lean mass only when compared to PWS control mice (18.47 g), and not to PWS-KO controls (19.0 g). Analysis of only Week 12 lean mass levels indicates a significant effect of genotype (*F* = 15.28, *p* < 0.001), with Tukey post hoc analysis showing WT lean mass (22.31 g) significantly higher than both PWS (19.36 g) and PWS-KO (19.16 g) lean mass (Figure 2C).

Body temperature readings were measured weekly over the entire study. Overall, the effect of genotype was not significant (*F* = 1.22, *p* = 0.29), and nor was treatment (*F* = 2.6, *p* = 0.11), but the effect of week was significant (*F* = 1.8, *p* = 0.038). However, there were no cross-interaction effects. Post hoc analysis using Student’s *t* to examine the overall effect of week, going from pre- to post-week data, showed that there was a significant reduction in overall body temperature from Week 1 to Week 12 (Week 1, 37.54 °C; Week 12, 37.20 °C, *p* = 0.03). However, by Week 12, body temperature showed no significant differences for genotype by treatment (*F* = 2.04, *p* = 0.14) (Figure 2D), while temperature over the whole study showed a significant effect of treatment (Appendix A).

Over the entire study, CLA diet showed a significant effect of treatment on food intake for genotype (*F* = 41.9, *p* <0.0001), treatment (*F* = 5.84, *p* = 0.016), and week (*F* = 9.97, *p* < 0.0001). Post hoc analysis of genotype indicated that WT mice ate approximately 2–3 g more food overall than PWS and PWS-KO mice (Appendix A), but this trend is changed when food intake is normalized to body weight. Then, PWS mice have a significantly higher overall food intake/gram body weight (Appendix A). CLA diet overall significantly increased food intake in all genotypes (16.48 g, control diet versus 17.18 g CLA diet). This increase for CLA diet intake was true even when food intake was normalized to body weight (Appendix A). When analyzing food intake only at the end of the study (Week 12), there was no overall significant difference between genotypes (*F* = 2.41, *p* = 0.10), treatments (*F* = 0.01, *p* = 0.98), or the interaction of genotype × treatment (*F* = 0.41, *p* = 0.66).

### 3.2. Changes in Fasting Glucose and Glucose Tolerance with Genotype and Treatment

In two previous studies, including one from our group, glucose tolerance was improved by treatment with the t9,11 isomers of CLA [16]. Glucose tolerance tests were performed both before diet treatment began (“Pre” Week) and at study end (Week 12). In the Pre Week, there was a significant effect of genotype on fasting glucose levels (*F* = 4.3, *p* = 0.0202), with post hoc tests revealing a significant increase in fasting glucose for PWS-KO mice over WT, but not PWS mice (Figure 3A). For the GTT curves, area-under-the-curve (AUC) measurements were significant for genotype (*F* = 4.19, *p* = 0.02), with the post hoc analysis showing significant differences between PWS and PWS-KO genotypes, but not WT mice (Figure 3B,C). Following diet treatment for 12 weeks, only fasting glucose levels showed a significant treatment effect (*F* = 7.5, *p* = 0.0092). Post hoc analysis revealed that fasting glucose for animals fed CLA diet (all genotypes) was significantly higher compared to control-diet-fed animals (Figure 3A). However, overall AUC levels were not significant for single or cross-effects (Figure 3D–F).

### 3.3. No Effect of CLA Treatment on Metabolism

According to a review of CLA effects on skeletal muscle metabolism, there are mixed effects on resting metabolic rate (RMR) in humans given CLA, although mice treated with CLA show consistently higher RMRs and respiratory quotient, a measure of fat oxidation [17]. TSE measures of metabolism were performed during Week 12 of the study. At this time, home-cage activity was also measured, and body composition for all mice was measured prior to putting them in TSE chambers so that energy expenditure per fat-free mass (lean mass) could be calculated. In this study, data from several mice could not be obtained during the Post-Week period due to a malfunction in our TSE system. Analysis of the available data (N between four and seven for each group) indicated there were no significant effects of genotype, treatment, and genotype × treatment on respiratory exchange ratio (RER) (VCO_2_/VO_2_) or on KJ per fat-free mass energy expenditure levels (Appendix A) (Figure 4). RER levels closer to 0.9 for all genotypes and treatments indicated higher fat oxidation, consistent with the HFD for the study (Figure 4A). There were slight but non-significant increases for energy expenditure for all genotypes with CLA diet, and these could be contributing at some level to the lower body weight and fat in the CLA-diet animals.

### 3.4. CLA Modulates Spontaneous Wheel Running Levels without Effects on End-of-Study Rotarod, Home-Cage Activity, or Anxiety Measurements

Voluntary physical activity can be increased in Nhlh2-knockout mice exposed to CLA diet [14]. We sought to determine if CLA diet would also increase voluntary and home-cage activity in PWS and PWS-KO mice on HFD. In addition, mice were tested on the rotarod to confirm balance and forward-motion abilities, and with an elevated plus apparatus to analyze anxiety-like behavior. Prior to diet introduction, there was a reduction in 24 h spontaneous wheel running for PWS-KO mice compared to WT, with the *p*-value just above the alpha of 0.05, (*F* = 3.18 *p* = 0.0510). Post hoc analysis demonstrated greater wheel revolutions for WT compared to PWS-KO mice in the pre-study period (Figure 5A). While PWS mice were not significantly different from either WT or PWS-KO mice, they did have an average reduction of 2103 revolutions per 24 h compared to WT mice (Figure 5A). Following the CLA intervention, the effect of genotype (*F* = 89.34, *p* = 0.0005; Appendix A) and the interaction of genotype × treatment (*F* = 4.08, *p* = 0.0242; Figure 5A) were significant, but not treatment alone (*F* = 0.94, *p* = 0.34; Appendix A). Post hoc analysis showed an interesting, significant increase in spontaneous running for WT mice on the control diet that was not found for PWS or PWS-KO mice. However, all three genotypes of mice on CLA diet had similar running wheel activity (Figure 5A). When only genotype is considered, the effect of CLA on home-cage activity, as measured while mice were in the TSE metabolic chamber at the end of the study, was not significant (Figure 5B). This could again have to do with the lower numbers of mice for each genotype and treatment group because of the TSE malfunction during the experimental time course, but is consistent with the 24 h wheel running levels with CLA treatment. As expected from the exercise findings, rotarod ability was not affected by genotype or CLA-diet treatment (Figure 5C). Finally, previous work detected increased anxiety in PWS mice on normal chow, by using the elevated plus apparatus to detect less time spent in the open arms of the apparatus [36,37]. In this study, mice were tested both pre- and post-intervention. However, while WT mice appeared to spend more time in the open arms than either the PWS or PWS-KO mice, there were no differences based on genotype, treatment, or the interaction of genotype × treatment (Figure 5D).

### 3.5. In Vivo Muscle Function Modulation by CLA Treatment

There are data suggesting that CLA can increase muscle strength in humans when combined with resistance training (e.g., [38]). At the end of the 12-week study, mice were subjected to an in vivo isometric plantarflexor torque–frequency assay to measure muscle strength in response to increasing frequency of electrical stimulation. As there were significant differences in body weight, but not lean mass of CLA groups, the data were normalized both to total body weight (Figure 6A) and lean body mass (Figure 6B).

Torque—frequency showed a significant effect of genotype regardless of whether it was normalized to total body weight or lean body weight (*F* = 16.42, *p* < 0.0001) or lean body mass (*F* = 7.063, *p* < 0.0001). When normalized to lean body mass or total body mass, torque was significantly reduced in WT mice on the control diet, compared to all other groups except PWS-KO (TBM *p* < 0.0001, LBM WT CLA, PWS *p* < 0.0001; LBM PWS-KO CLA *p* = 0.0020). Interestingly, CLA treatment increased torque output in WT mice, regardless of normalization method (*p* < 0.0001). PWS mice on the control diet showed a significant increase in torque over PWS mice on the CLA diet when data were normalized to lean mass (*p* < 0.0336), but not when total body mass was used. (*p* = 0.9993). Resistance to fatigue was greatest in the PWS-KO group (PWS *p* = 0.0002, other groups *p* < 0.0001). WT (PWS, PWS-KO *p* < 0.0001, PWS-KO CLA *p* = 0.0172) and PWS (PWS, PWS-KO *p* < 0.0001) mice on CLA diet both showed greater fatigue than most other genotypes and treatments (Figure 6C). CLA treatment was not detrimental to muscle function in any group, as all metrics remained greater than or equal to the WT group, which serves as a baseline.

### 3.6. CLA-Treatment Effects on Liver

Multiple studies have shown that, in addition to fat loss, a side-effect of CLA treatment is liver steatosis, which is even more prominent with the t10,12 isomer of CLA as compared to the t9,11 isomer [39]. As Tonalin™ CLA contains a 50:50 mixture of these isomers, a finding of liver steatosis, or fatty liver, in CLA-treated mice from all genotypes was not unexpected (Figure 7). However, as shown, while all liver sections from WT mice with CLA diet showed steatosis, only five of eight PWS mice on the CLA diet showed signs of steatosis in their livers—and two of these were mild, while four of seven PWS-KO mice on CLA diet had visible steatosis (compare Figure 7E with Figure 7F, and Figure 7G with Figure 7H).

### 3.7. RNA-Seq Analysis of Hypothalamic Gene Expression

NHLH2 and SNORD116 are both highly expressed in the adult hypothalamus [40,41]. As hypothalamic dysfunction is highly implicated in PWS, this study sought to understand the effects of CLA on the PWS and WT mouse hypothalamus. RNA was isolated from dissected hypothalamic blocks to identify differentially regulated genes that may be effectors of the genotype or diet. As shown in Figure 8A, a total of 431 differentially expressed genes were found between control diet and CLA diets, while only 35 differentially expressed genes were detected when WT and PWS mice were compared. PWS-KO mice were not used in these analyses. The magnitudes of differential expression were modest for all conditions, with nearly all differentially expressed genes showing fold-changes between 0.5 and 2 (Date Appendix A), although edgeR calculated higher fold-changes. Necdin (Ndn), a gene whose expression is increased by Nhlh2 [42] and whose gene locus lies within the PWS Type I (large) deletion on human chromosome 15q [9,43], was increased in PWS animals, a finding that has been reported previously [44]. Interestingly, there was no differential expression of Nhlh2 mRNA for this study, even though our laboratory has shown it to be post-transcriptionally regulated by Snord116 [5]. Of all differentially expressed genes identified by RNA-seq, 135 were up-regulated by CLA diet and 22 were upregulated in the PWS genotype. Interestingly, one gene was “rescued” by CLA treatment in PWS mice, *Mael,* in that it is expressed in WT mice, and not in PWS except when they are treated by CLA. Additionally, the *Dgkk* gene was found to be overexpressed in PWS, consistent with recent in vitro findings [45]. However, these findings in *Mael* and *Dgkk* have yet to be confirmed by RT-QPCR. Gene Ontology (GO) analysis (Appendix A) revealed that CLA diet treatment resulted in significantly enriched processes related to ribosomal RNA processing, cellular metabolic processes, and aging, lethality, or survival processes (Appendix A). When only the WT response to CLA diet was examined, starvation, nutrient levels, and cellular stress were significantly enriched (Appendix A). Only two pathways, RBM3/CIRBP RNA recognition, and abnormal bone mineral content were enriched when a genotype comparison of PWS versus WT was analyzed (Appendix A). Four differentially expressed genes were chosen using both biological function, reproducibility, and magnitude of change. These genes underwent further confirmation validation by QPCR for all diet and genotype conditions, with only two of the four findings replicated. As shown in Figure 8B, the glutamate receptor, ionotropic, AMPA4 subunit (Gria4) mRNA, which was 1.23-fold up-regulated in PWS mice compared to WT mice in the original RNA-seq data, was not found to be significantly increased with QPCR (*p* = 0.07). However, the gamma-aminobutyric acid (GABA) A receptor, subunit gamma 2 (Gabrg2) mRNA maintained its significant increase in PWS mice compared to WT mice, although the increase of 1.16-fold by QPCR is modest (Figure 8B; *p* = 0.02). In comparing the effect of diets on both genotypes, charged multivesicular body protein 1B (Chmp1b) mRNA was 1.27-fold increased with CLA diet for RNA-seq, and this increase was maintained in the verification QPCR results (Figure 8C, *p* = 0.01). Reticulophagy regulator family member 2 (Retreg2) showed a significant 0.89-fold reduction in the RNA-seq analysis of CLA diet treatment on PWS mice, but this modest reduction was not significant in the validation QPCR results (Figure 8C; *p* = 0.22).

### 3.8. Cecum Microbiome Analysis in CLA and Control Diet-Treated Animals

Using a mix of isomers similar to those used in our study, Li and colleagues demonstrated a significant reduction in pro-inflammatory bacteria in CLA-fed high-fat-diet mice [46]. Other studies using human patients with multiple sclerosis have also been able to demonstrate an anti-inflammatory microbiome profile with CLA treatment [47]. Therefore, we set out to determine if bacterial diversity and specific genera and species were similar or different in PWS and WT mice on CLA or control diet (Figure 9).

α-diversity describes the structure of a microbial community in relation to the number of taxonomic groups and their respective abundance. α-diversity metrics were similar by genotype (*p* ≥ 0.58) but differed between the control and CLA treatments (*p* ≤ 0.02; Appendix A). β-diversity represents community compositional differences between samples. Here, as shown in Figure 9A, β-diversity measures were significantly impacted by CLA diet (*p* > 0.05), whereas genotype had no effect. At phylum level, the relative abundance of *Cyanobacteria* was lower in mice fed the CLA diet compared to those fed the control diet (*p* = 0.04), whereas the relative abundance of the other phyla was similar (*p* > 0.09) between diets. Amongst genotypes, PWS-KO mice had significantly lower relative abundance of *Bacteroidetes* compared to WT mice (*p* = 0.04), but similar compared to PWS (*p* = 0.26). Reduced abundance of *Bacteroidetes* phylum is generally associated with obesity [48]. For *Proteobacteria*, PWS mice had a lower relative abundance (*p* = 0.05) compared to WT mice, but similar compared to PWS-KO (*p* = 0.73) (Figure 9B). Only three significant genera were identified in this study (Figure 9C), with all three being impacted by diet. *Ruminococcus sp.* Is important for plant cell wall breakdown in the colon [49], and in our study, was reduced with CLA treatment for all genotypes. While not much is known about *Sutterella sp.* and its role in gut microbiome health, the genus was significantly increased with CLA diet (*H* = 13.8, *p* = 0.02). *Turicibacter* is a genus in the *Firmicutes* phylum, with this phylum generally making up the largest portion of the gut microbiome [50]. CLA treatment significantly reduced *Turicibacter sp.* In all genotypes (*H* = 15.7, *p* = 0.008), although the *Firmicutes* phylum overall was not significantly affected (*H* = 3.74, *p* = 0.59).

## 4. Discussion

Obesity and the complications of increased body weight and fat mass represent one of the main adult phenotypes impacting health and well-being for patients with PWS. The main goal of the study was to determine if CLA treatment would result in body weight and fat loss in mice with a single or double deletion of *Snord116* (PWS, or PWS-KO mice, respectively). The *Snord116* locus encodes multiple small nucleolar RNAs and is one of the three genes within the minimal causative genomic region for PWS [43]. Mice with a single paternal deletion of *Snord116,* as well as a deletion of both alleles of *Snord116*, were included in this study. Our findings demonstrate that deletion of *Snord116* either hemi- or homozygously does not interfere with the weight- and fat-reducing properties of CLA. Despite the lack of overt obesity of the PWS and PWS-KO mice, even on a 20% fat diet, both genotypes showed significant body weight loss due to body fat and not lean (muscle) mass loss. Of note, Tonalin™, which is a 50:50 mixture of CLA isoform, has mixed results overall in weight reduction for humans, perhaps due to the wide variability in dosage per gram body mass given in each study [17]. In humans, studies achieving weight loss were dosing at a minimum of 2.8 g/day/kg body mass for at least 4 months. In this study, a Tonalin™ CLA dosing was calculated to be at an equivalent dose of 5.2 g CLA/kg body mass in free-feeding animals (0.13 g total per day). A study of 73 patients (age range 16–58 years), with either deletion of the locus (64%) or uniparental disomy (36%), the average body mass was 99.4 kg and 81.0 kg, respectively, and in the overweight-to-obese range for those individuals [51]. Thus, for an adult individual with PWS at ~90 kg, a daily dosage of 468 g of CLA would be equivalent in grams CLA/kg body mass. Obviously, this is not achievable or within GRAS guidelines, so dosage of patients with PWS would have to be titrated to find the optimal range for body weight loss.

Patients with PWS experience increased body weight due to both hyperphagia and reduced physical activity [43,52,53]. There is also a documented reduction in overall energy expenditure in these patients [52]. In examining the possible mechanisms of action for CLA treatment, there does not seem to be any reduction in food intake with CLA intake, either in this study or in other reports in both mice (e.g., [54]) and in overweight humans (e.g., [55]), although results are mixed (e.g., [56]). In the study by Kamphius and colleagues [55], feelings of fullness and satiety were increased and hunger was decreased with CLA supplementation. This could be an important consideration for patients, perhaps giving the ability to eat regularly, rather than diet, or could work in conjunction with a low-fat, low-calorie diet regimen. Habitual physical activity levels are significantly lower in patients with PWS compared to both controls with and without obesity [57]. It was anticipated that, like N2KO mice treated with CLA [14,15,16], PWS mice would demonstrate increased 24 h spontaneous wheel running with CLA treatment compared to control. However, while there was a significant effect of genotype on running wheel activity prior to the intervention, there were no differences in genotype post-treatment. Likewise, home-cage activity, as measured in the TSE apparatus, was not different for the effect of either genotype or treatment.

Patients with PWS display overall muscle hypotonia and functional weakness. Several studies report reduced cross-sectional area and unusual morphology of PWS-affected muscles [58,59]. Few studies have quantified muscle strength in PWS patients [59,60]. Females with PWS showed lower-knee extensor and flexor torque than both obese and non-obese controls [60]. When plantarflexor muscle function was measured in adults with PWS, lower absolute peak torque output was produced than controls with obesity [59]. When normalized to cross-sectional area, there were no differences in torque between PWS patients and controls [59]. Similarly, the present study demonstrates that PWS and PWS-KO mice do not display plantarflexor weakness, with or without CLA treatment. In healthy subjects, CLA supplementation in combination with resistance training has been reported to better maintain lean body mass and improve some metrics of strength in humans [38]. Similarly, a study in older WT mice fed a high-fat diet and supplemented with a combination of CLA and omega-3 increased grip strength and improved body composition after performing resistance training [61]. However, a control group from the same study that did not perform resistance training but did receive CLA and omega-3 supplementation displayed lowered grip strength and cross-sectional area of muscle [61]. Our data indicate that CLA supplementation was beneficial to torque output in WT mice. In addition, PWS and PWS-KO mice did not show decreased torque output in comparison to WT mice, regardless of normalization type or CLA treatment. CLA treatment lowered torque output in the PWS group normalized to lean body mass. Resistance to fatigue was also lowered in PWS CLA and PWS-KO CLA groups compared to their control diet counterparts. However, CLA does not lower any of the torque measurements below WT. Therefore, CLA is not detrimental to muscle function, but does provide favorable changes in body composition.

Several studies have demonstrated that CLA can cause browning of white adipose tissue (e.g., [62]). As brown adipose tissue is important for heat generation, this would be expected to increase body temperature. Interestingly, in our mice on the CLA diet, body temperature dropped significantly regardless of genotype, and this is consistent with another study where mice on CLA diet exposed to cold had significant drops in body temperature compared to control mice [63]. Thus, it appears that body temperature may be more responsive to the loss of insulation from the loss of white adipose than any potential browning of the remaining white adipose in the CLA-treated mice. Metabolic analysis revealed that both respiratory exchange ratio and metabolic rate were unaffected by genotype or treatment, although there did appear to be a higher metabolic rate with CLA pre- versus post-treatment, similar to that seen in mice treated with CLA (e.g., [64]) and in some human studies ([17]). As there was a failure of our TSE system during the study, it is possible that our data would have shown significant increases in metabolic rate if all animals in the study had been included in the metabolism analysis. These data suggest that CLA treatment is having some non-food intake, non-exercise effect on body weight, consistent with the increased (although non-significant) effect on whole-body energy expenditure. Any future study of CLA treatment of PWS patients in humans should consider overall body metabolism analysis to be an important measurement.

In further attempting to characterize physiological differences in CLA-treated PWS and PWS-KO mice, fasting glucose and glucose tolerance tests, as well as liver histology, were measured. Not unexpectedly, CLA treatment significantly increased fasting glucose, and overall AUC in GTT tests, regardless of genotype. Likewise, CLA treatment increases the incidence of liver steatosis regardless of genotype. Several reports have demonstrated that it is the t10, c12 CLA isomer that is responsible for liver steatosis development, and that the mechanism appears to be due to increased circulating free fatty acids and glucose that are picked up by the liver, and not adequately oxidized [65]. Proteomic and molecular analyses of fatty liver in mice treated with CLA have shown increases in gene expression and levels of proteins related to lipogenesis [65,66]. In our study, 100% of WT mice demonstrated liver steatosis on CLA diet, but only ~50% of PWS and PWS-KO mice did. While there is mixed agreement as to whether CLA would have similar effects on hepatic steatosis in humans [17], our results demonstrate that the PWS genotype may protect against this condition, but more work would be needed to understand the mechanisms occurring.

We undertook an RNA-seq approach to attempt to identify Snord116-deletion-specific effects and CLA-specific effects on mRNA expression in the hypothalamus, possibly linking these to relevant pathways. This is the first RNA-seq study examining the hypothalamus in mice with a CLA-supplemented diet, to the authors’ knowledge [5,6,14,15,16,40,43]. In total, 580 genes were differentially expressed using an FDR of 0.10 and most of these only had modestly changed RNA levels between 0.5- and 2-fold. The gene with the most differential expression was Ipw, a non-coding RNA gene that is deleted in both PWS patients and the PWS mouse [43], so this finding was expected and confirmed the overall differential expression analysis. Significant gene ontology pathways for control diet versus CLA diet of nutrient and stress responses were unsurprising but may still yield future information on the mechanism of action of CLA in the hypothalamus. Interestingly, one gene, *Mael*, was “rescued” by CLA in PWS mice. *Mael* plays a central role in spermatogenesis [67], but the overall expression level in hypothalamus was very low, making it difficult to assess any biological relevance of this finding. Only one gene was confirmed to be differentially regulated between control and CLA treatments (all genotypes) by secondary qPCR analysis, *Chmp1b*. The protein product of *Chmp1b* is an endosomal-associated protein involved in the endosomal sorting complexes required for transport (ESCRT) that may be involved in the multi-vesicular body (MVB), a specialized endosome [68,69,70]. Chmp1b protein has been implicated to support lipid-droplet to peroxisome fatty-acid trafficking [71]. As *Chmp1b* was increased with CLA diet, these data suggest there may be more membrane protein turnover or fatty-acid oxidation with CLA. However, more studies are needed to confirm this finding. The RBM3/CIRBP RNA binding protein GO pathway, which was significant for WT versus PWS mice, includes *Rbm3* and *Cirbp*, which are two RNA binding proteins induced by hypothermia, and modulated during circadian rhythms [72]. This is an interesting finding as both PWS mice and individuals with PWS have altered circadian rhythms and sleep cycles [73]. The two RNA binding proteins are thought to control polyadenylation of mRNAs, changing post-transcriptional mRNA stability. This finding links our more recent finding that *Nhlh2* mRNA stability is modulated by *Snord116,* possibly through a motif that is very near to the end of the transcript [5], as well as our previous data showing *Nhlh2* mRNA oscillation with cold-temperature exposure [74]. Future studies to examine this pathway may help to clarify any relationship between downregulation of *Nhlh2* in PWS and hypothalamic circadian and sleep patterns. Finally, for WT versus PWS mice (all treatments) two genes were further analyzed by qPCR, but only one, the *Gagbr2* gene encoding GABA receptor Type A, subunit gamma2, was significantly upregulated in PWS mice with secondary analysis. In the PVN of the hypothalamus, Gabrg2 protein has been implicated in diurnal rhythmicity in metabolism and diet-induced obesity through upstream regulation by the circadian gene Bmal1 [75]. Loss of Gabrg2 in the PVN leads to obesity and loss of diurnal rhythm of energy expenditure and food intake [75]. Additionally, Gabrg2 is diurnally regulated in the suprachiasmatic nucleus (SCN) of hamster and mice [76]. Interestingly, GABA neurotransmitter levels are reduced in individuals with PWS, and it is thought that this reduction can lead to some of the neurodevelopmental and behavioral problems with those patients [77]. Upregulation of one of the receptors for GABA, a major neuro-inhibitory neurotransmitter could be secondary to the reduced GABA brain levels, possibly contributing to the overall GABAergic dysfunction. Interestingly, in patients who have PWS as a result of the large deletion on 15q, other forms of the GABA A receptor can be included in the deletion [78]. Of note, we failed to find differential expression of some genes implicated in hypothalamus dysfunction of PWS mice, including *Nhlh2*, *Pomc*, *Npy*, and others. We hypothesize that a cell-type specific approach under specific signaling conditions may better capture sensitivity to Snord116-mediated regulation. Overall, the mRNA-seq and follow-up RT-QPCR show very modest changes, if any. The weak effect size for all differentially expressed genes in the current RNA-seq study also leads to difficulty in validation of differentially expressed genes, as RT-QPCR is limited by the sample size in the current study and their natural variability. Similarly, previous RNA expression studies of hypothalamus tissue from PWS mouse models have shown modest effect sizes [8,11,79,80,81].

As this was a dietary intervention, microbiome analysis was performed to determine if CLA diet, the PWS (*Snord116^m+/p−^*) genotype, or the interaction of these could have effects on gut microbiota. Previous analyses of individuals with PWS have found that those individuals with obesity share a similar microbiome compared to individuals with non-syndromic obesity [82,83], and that a dietary intervention that modulated obesity had similar effects on the microbiome, namely, an increase in *Bifidobacterium* sp. [83]. PWS and PWS-KO mice also shared a similar microbiome with WT mice (i.e., no genotype differences). These findings are somewhat in contrast to a 2021 study showing that individuals with PWS had significantly fewer *Bifidobacterium* sp. than others in the study, including individuals with irritable bowel syndrome [84]. However, in that same study, levels of *Tenericutes* were significantly higher in those individuals with PWS, which was not replicated in our mouse model. *Turicibacter* sp. decreased in CLA diet, and while not much is published on this genus, it was also found to be low in Type I diabetic children [85]. Relative abundance of *Bacteriodetes* and *Firmicutes* showed the trend towards decrease and increase (non-significant), respectively, in PWS and PWS-KO compared to WT mice. This is consistent with overall *Bacteriodetes*/*Firmicutes* profile in obese vs. normal-weight individuals. Firmicutes are known for their energy harvesting capabilities and tend to show increase in obese individuals [86]. The microbiome field has been explored to great length in many studies of late, however, the consensus on its being causative or merely associative is still debated in many of them. In the current study, the modest changes we see in microbiome composition in the PWS model, are not per se causative, but show an association. Before we can place microbiome as one of the mechanisms, microbiome-derived metabolites, secondary bile acids, and the whole metabolome need to be studied. However, the microbiome aspect of the current study opens up a new area of investigation in the unique mouse models we used in this study.

## 5. Conclusions

Overall, the original hypothesis that CLA diet would reduce body weight and body fat in mice carrying a paternally inherited deletion of *Snord116* was confirmed. Consistent with the literature on CLA diet, there was no detectable change in food intake throughout the 12-week study. While fasting glucose was, as expected, increased with CLA diet in all genotypes, there was an unexpected difference in hepatic steatosis, with about half of the PWS and PWS-KO genotype mice not showing fatty liver, compared to all of the WT mice on CLA diet. This difference appears to be genotype-specific, although more work is needed to determine the mechanism. Interestingly, neither lean mass nor muscle function were compromised or improved by CLA treatment. Metabolism was also unchanged, leading us to be unable to conclude the mechanism for CLA-induced body fat loss. Studies that were designed to tease out CLA-induced or -suppressed pathways through either mRNA regulation in the hypothalamus or microbiome analysis in the gut have provided intriguing leads for future studies.

## 6. Patents

VTIP 21-068: Reversal of Prader–Willi Syndrome Symptoms with Tonalin Conjugated Linoleic Acid, U.S. Provisional Application No. 63/184,001, to D.J.G.

## Figures and Tables

**Figure 1 nutrients-14-00860-f001:**
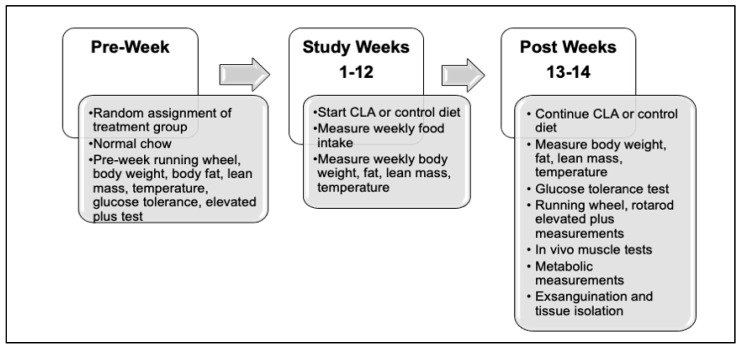
Experimental study design. Mice were randomly assigned to their treatment group based on genotype over the 1.5 years of the study. Following study baseline measures during the pre-week period, the standard chow diet was changed to study diet and continued for up to 14 weeks or until euthanasia.

**Figure 2 nutrients-14-00860-f002:**
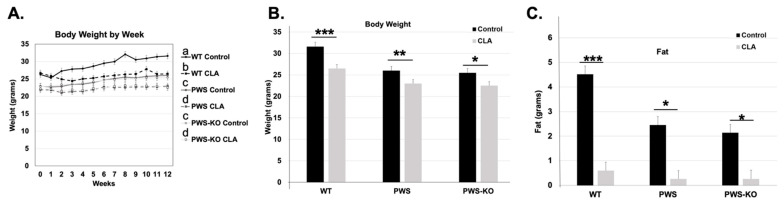
Overall CLA study effects on Week 12 weight, fat, lean mass, temperature, and food intake. Post hoc analysis findings for (**A**,**B**) body weight, (**C**) fat, (**D**), lean mass, (**E**) temperature, and (**F**) food intake, from Week 12 data points for the 12-week study. All data are presented as mean +/− standard error of the mean. *N* = 8 WT control, 8 WT CLA, 8 PWS control, 8 PWS CLA, 7 PWS-KO control, 8 PWS-KO CLA. *** *p* < 0.001, ** *p* < 0.01, * *p* < 0.05. Individual statistics for each measure are provided in text, and as data tables in Appendix A tables. WT = (*Snord116^m+/p+^*), PWS = (*Snord116^m+/p−^*), PWS-KO = (*Snord116^m−/p−^*).

**Figure 3 nutrients-14-00860-f003:**
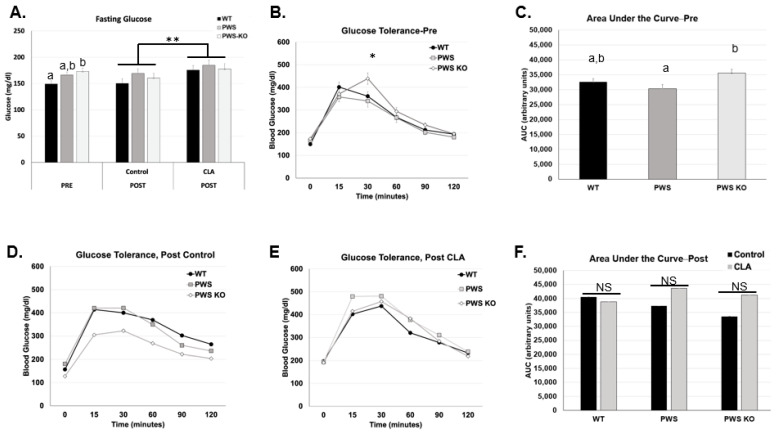
Fasting glucose and glucose tolerance findings with CLA diet. Glucose measurements were performed in the week prior to diet treatment and at the end of the study (12 weeks). Post hoc analysis findings for fasting glucose (**A**) for both study times, (**B**) glucose tolerance for each genotype pre-study, (**C**) area-under-the-curve values for each genotype, pre-study, (**D**) glucose tolerance curves for each genotype on control diet, (**E**) glucose tolerance curves for each genotype on CLA diet, and (**F**) area-under-the-curve values for each genotype and treatment group. All data are presented as mean +/− standard error of the mean. *N* = 8 WT control, 8 WT CLA, 8 PWS control, 7 PWS CLA, 7 PWS-KO control, 8 PWS-KO CLA. Letters indicate significant differences within a single figure for effects of genotype. For effect of treatment, * indicates an effect of treatment at the *p* < 0.05 level, ** indicates an effect of treatment at the *p* < 0.01 levels, and NS indicates the differences are not significant. Individual statistics for each measure are provided in text, and as data tables in Appendix A. WT = (*Snord116^m+/p+^*), PWS = (*Snord116^m+/p−^*), PWS-KO = (*Snord116^m−/p−^*).

**Figure 4 nutrients-14-00860-f004:**
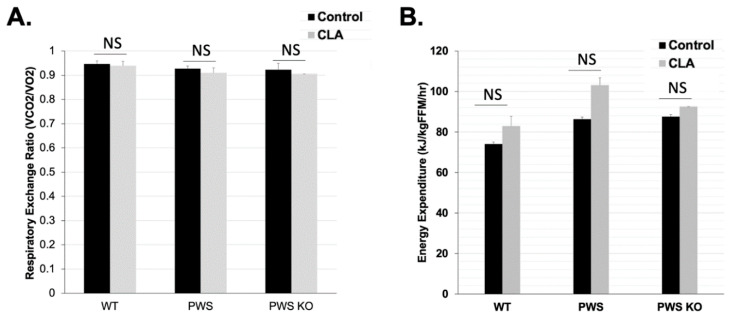
Metabolic rate in control and CLA-treated animals. TSE system measurements were performed at the end of the 12-week study. Effect tests of genotype, treatment, and genotype × treatment were all non-significant. (**A**) Respiratory Exchange Ratio (VCO_2_/VO_2_) and (**B**) Energy Expenditure (KJ/kg/FFM/h). Due to a broken/malfunctioning part that occurred during the study, only the following animal numbers from the entire study were tested in the TSE system: *N* = 7 WT control, *N* = 5 WT CLA, *N* = 6 PWS control, *N* = 6 PWS CLA, *N* = 4 PWS-KO control, *N* = 4 PWS-KO CLA. Individual statistics for each measure are provided in text, and as data tables in Appendix A. WT = (*Snord116^m+/p+^*), PWS = (*Snord116^m+/p−^*), PWS-KO = (*Snord116^m−/p−^*).

**Figure 5 nutrients-14-00860-f005:**
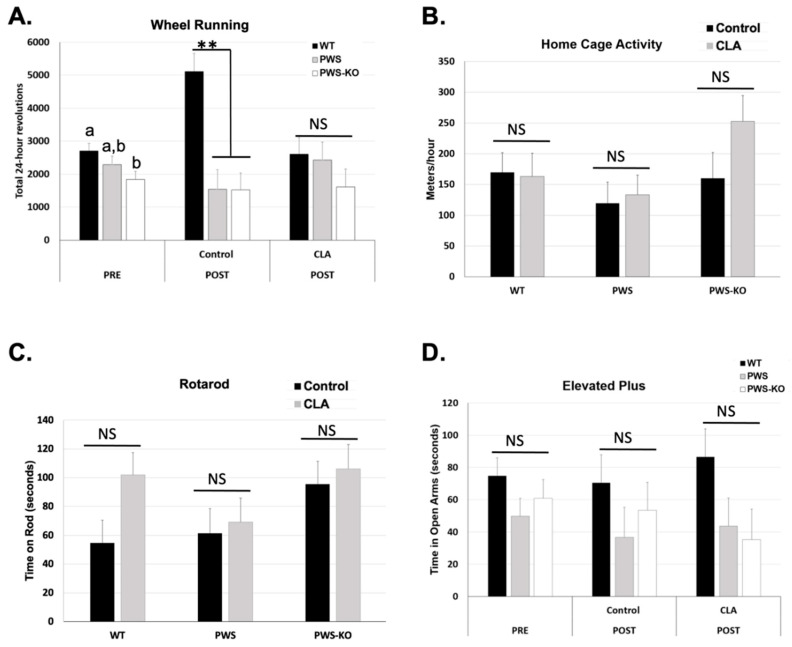
Activity measures in control and CLA-treated mice. (**A**) Mice were placed in cages containing running wheels for 72 h, with the first 24 h of running discounted for acclimation to the new cage. Wheel running activity was measured in both the pre- and post-study time periods. Twenty-four-hour measurements were collected. *N* = 8 WT control, *N* = 8 WT CLA, *N* = 7 PWS control, *N* = 6 PWS CLA, *N* = 8 PWS-KO control, *N* = 7 PWS-KO CLA. (**B**) Home-cage activity in the X–Y–Z axes was collected via beam breaks during the calorimetry measurement in the post-study period. The software converts these readings to meters per hour. *N* = 7 WT control, *N* = 5 WT CLA, *N* = 6 PWS control, *N* = 6 PWS CLA, *N* = 4 PWS-KO control, *N* = 4 PWS-KO CLA. (**C**) Following three days of rotarod acclimation, fourth-day rotarod tests were conducted in the post-study period. The average of four tests for each animal is shown. *N* = 8 WT control, *N* = 8 WT CLA, *N* = 7 PWS control, *N* = 6 PWS CLA, *N* = 8 PWS-KO control, *N* = 7 PWS-KO CLA. (**D**) Time in open arms was determined with a 5 min testing period. *N* = 8 WT control, *N* = 8 WT CLA, *N* = 6 PWS control, *N* = 6 PWS CLA, *N* = 8 PWS-KO control, *N* = 6 PWS-KO CLA, NS-non-significant. For effect of treatment, ** indicates an effect of treatment at the *p* < 0.01 level. Individual statistics for each measure are provided in text, and as data tables in Appendix A. WT = (*Snord116^m+/p+^*), PWS = (*Snord116^m+/p−^*), PWS-KO = (*Snord116^m−/p−^*).

**Figure 6 nutrients-14-00860-f006:**
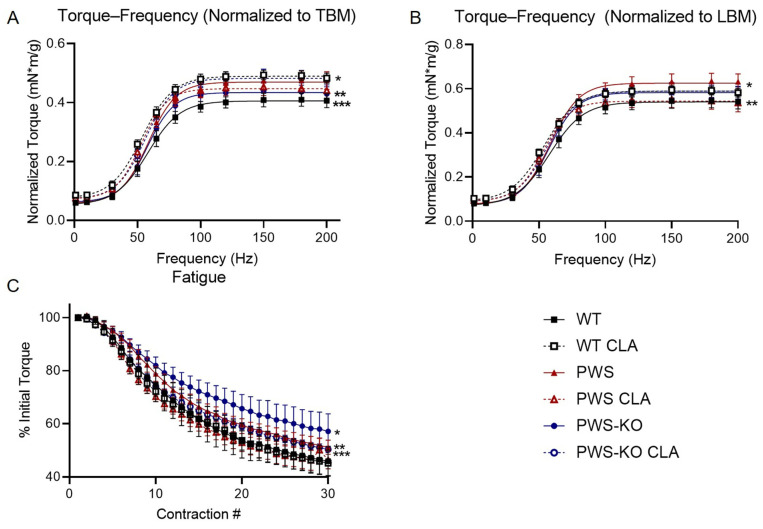
In vivo plantarflexor torque measurements. (**A**) Torque frequency, normalized to total body mass (TBM). * WT CLA > PWS, PWS CLA, PWS-KO. ** PWS-KO < PWS-KO CLA. *** WT < WT CLA, PWS, PWS CLA, PWS-KO CLA. (**B**) Torque-frequency, normalized to Week 12 lean body mass (LBM) to account for differences in total body weight. * PWS > PWS CLA. ** WT < WT CLA, PWS, PWS-KO CLA. (**C**) Fatigue as percent of initial contraction peak. * PWS-KO > all other groups. ** PWS > WT, WT CLA, PWS CLA. *** PWS-KO CLA > WT CLA, PWS CLA. Comparisons *p* < 0.05. *N* = 8 WT control, *N* = 8 WT CLA, *N* = 8 PWS control, *N* = 7 PWS CLA, *N* = 8 PWS-KO control, *N* = 7 PWS-KO CLA. WT = (*Snord116^m+/p+^*), PWS = (*Snord116^m+/p−^*), PWS-KO = (*Snord116^m−/p−^*).

**Figure 7 nutrients-14-00860-f007:**
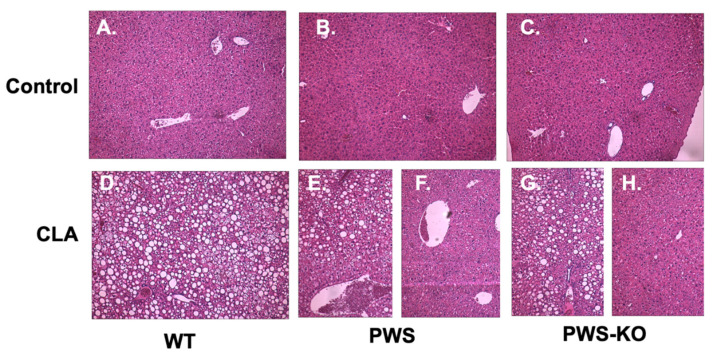
Liver histology in control and CLA-diet-treated mice. Liver biopsies were obtained from all mice that completed the study at necropsy. *N* = 3 samples from each genotype and treatment group were prepared for histological analysis with H&E staining. Representatives from all samples examined are shown at 10× magnification. (**A**) WT control diet, *N* = 8; (**B**) PWS control diet, *N* = 8; (**C**) PWS-KO control diet, *N* = 7; (**D**) WT CLA diet, *N* = 8; (**E**) PWS CLA diet showing liver steatosis, *N* = 5; (**F**) PWS CLA diet showing normal liver morphology, *N* = 3; (**G**) PWS-KO CLA diet showing liver steatosis, *N* = 4; (**H**) PWS-KO CLA diet showing normal liver morphology, *N* = 3. WT = (*Snord116^m+/p+^*), PWS = (*Snord116^m+/p−^*), PWS-KO = (*Snord116^m−/p−^*).

**Figure 8 nutrients-14-00860-f008:**
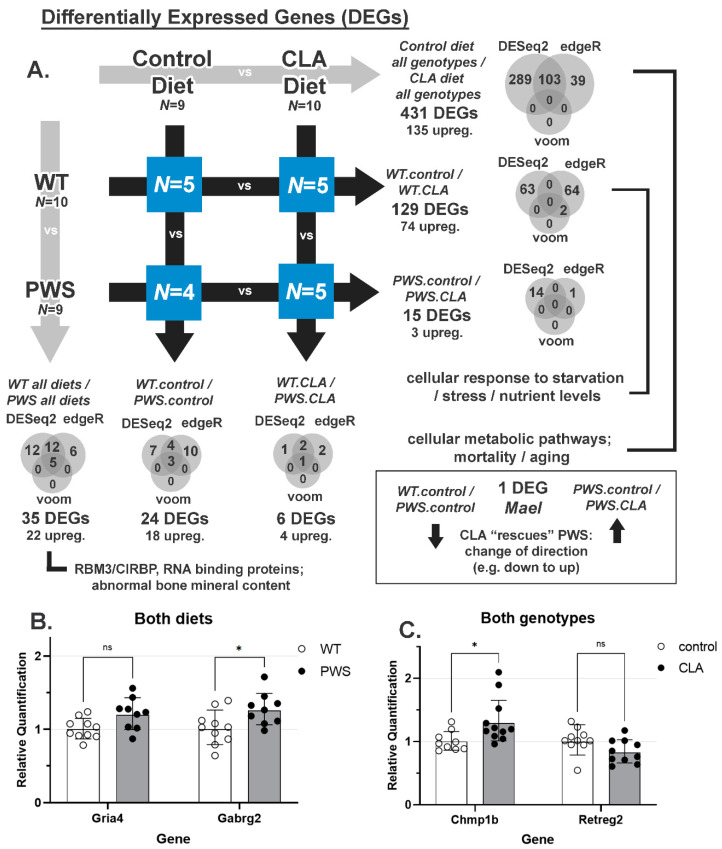
RNA-seq analysis and results in CLA-treated and untreated WT and PWS animals. At the end of the study, hypothalamic RNA was isolated from PWS and WT animals from both diet groups. RNA-seq analysis was performed on *N* = 5 (WT, control and CLA diet, PWS CLA diet) and *N* = 4 (PWS control diet). (**A**) Differentially regulated genes (DEGs) in all possible cross-conditions. All differentially expressed genes are listed in Appendix A. Significant GO terms are indicated (Appendix A). (**B**) Two genes that were differentially regulated between WT and PWS mice, regardless of diet, are shown. (**C**) Two genes that were differentially regulated with diet (no genotype effect) are shown. ns = non-significant; * = *p* < 0.05. *N* = 9–11 samples for QPCR per analysis. Graphs show geometric mean ± geometric SD, normalized to WT animals (**B**) or control diets (**C**). WT = (*Snord116^m+/p+^*), PWS = (*Snord116^m+/p−^*), PWS-KO = (*Snord116^m−/p−^*).

**Figure 9 nutrients-14-00860-f009:**
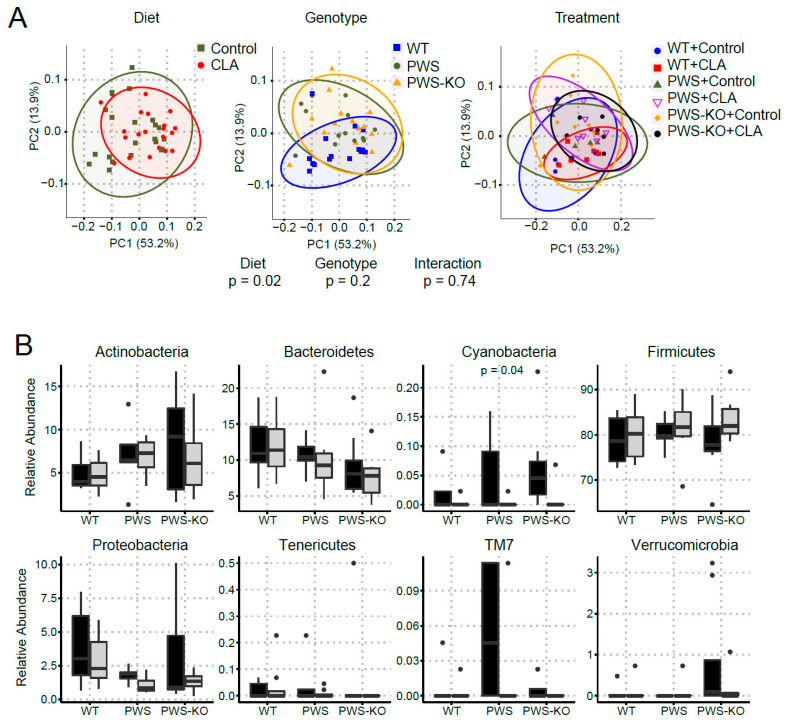
Changes in bacterial diversity in mice fed control and CLA diets. DNA isolated from cecal contents was used for 16S rRNA sequencing. (**A**) Principal coordinate analysis (PCoA) displayed no clustering of the bacterial communities by treatment or genotype; however, bacterial communities of mice clustered by diet. (**B**) At the phyla level, the relative abundance of Cyanobacteria differed by diet (*p* = 0.04). (**C**) At genus level, the relative abundance of Ruminococcus (*p* = 0.02), Sutterela (*p* = 0.01), and Turicibacter (*p* = 0.007) differed by diet. The Kruskal-Wallis test was used to determine the overall effects of diet, genotype, and their interaction. WT = (*Snord116^m+/p+^*), PWS = (*Snord116^m+/p−^*), PWS-KO = (*Snord116^m−/p−^*).

**Table 1 nutrients-14-00860-t001:** Experimental treatment groups.

Genotype	Treatment 20% Fat Diet	Treatment 20% Fat Diet with CLA
*WT*	8	8
PWS (*Snord116^m+/p−^*)	8	8
PWS-KO (*Snord116^m−/p−^*)	7	8

## Data Availability

Raw sequencing data for the mRNA-seq analysis are openly available at NCBI’s SRA database under BioProject PRJNA798246, scheduled for public availability on 2 February 2022.

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
