# Peer review of "Dietary Conjugated Linoleic Acid Reduces Body Weight and Fat in Snord116m+/p− and Snord116m−/p− Mouse Models of Prader–Willi Syndrome"

_nutrients, 2022, doi:10.3390/nu14040860_

Round 1
Reviewer 1 Report
Overall, this is a well conducted study, with truthful interpretation of the findings. Unfortunately, the use of current PWS mouse models (Snord116del and PWScr (PMID: 18320030; PMID: 18166085)) to study the biogenesis of obesity is questionable, because apart from growth retardation, these models does not recapitulate the late stages of the human PWS phenotype (i.e. morbid obesity, metabolic syndrome etc.). Indeed, the authors found no obesity in PWS mice or striking differences in CLA consumption between the WT and PWS genotype.
Recently, based on data obtained from transfected cells the authors suggested that Snord116 could directly interact with processed Nhlh2 mRNA (3’UTR) and (post-transcriptionally) increase its stability. Although based on experimental data it was tempting to suggest this hypothesis; Snord116 is a C/D box snoRNA that exhibits nucleolar localization and is not detected in the cytoplasm under physiological conditions in vivo. Hence, it is no surprise that the authors did not observe any differences in the expression levels of Nhlh2 mRNA between WT and PWS hypothalami; questioning the biological relevance of previous cell culture transfection experiments. However, to my surprise, knowing that there is no effect on hypothalamic expression of Nhlh2 mRNA after Snord116 elimination in vivo, the authors continued to promote and speculate on Nhlh2 mRNA - Snord116 interaction and wrote the manuscript in this context (here I would like to remind about the lesson than we learned about pseudo-involvement of Snord115 in alternative splicing of 5HT2c pre-mRNA (PMID: 16357227; PMID: 33016258)). (*Comment not relevant to this manuscript: I would recommend the authors to check the Snord116 processing from the expression construct using Northern blot (PMID: 30124848)).
CLA and its effect has been studied for decades now, in various animal models, for example:
mice:
https://pubmed.ncbi.nlm.nih.gov/9270977/
https://pubmed.ncbi.nlm.nih.gov/10230716/
https://pubmed.ncbi.nlm.nih.gov/9728060/
https://pubmed.ncbi.nlm.nih.gov/19423318/
https://pubmed.ncbi.nlm.nih.gov/20180981/
https://pubmed.ncbi.nlm.nih.gov/28245284/
rats:
https://pubmed.ncbi.nlm.nih.gov/10827208/
hamster:
https://pubmed.ncbi.nlm.nih.gov/10613761/
pigs:
https://pubmed.ncbi.nlm.nih.gov/12575906/
as well as in humans (adults as well as children):
https://pubmed.ncbi.nlm.nih.gov/11110851/
https://pubmed.ncbi.nlm.nih.gov/15159244/
https://pubmed.ncbi.nlm.nih.gov/20200257/
etc……
In general, there are a number of studies, placebo controlled, double-blinded etc. which investigated the effects of CLA in overweight and obese mammals (including humans). However, as mentioned above, the PWS mouse model chosen does not exhibit any of these phenotypes, which raises questions about the overall aim of the study.
An unexpected finding in my option is observed differences between PWS and PWS-KO mice. Because both mice harbour a paternal deletion, they equally do not express ncRNAs from the PWS locus (leaking Snord116 expression from the maternal chromosome has not been reported for C57BL/6 genetic backgrounds). Maternal deletion does not affect expression of the imprinted PWS-locus. Could it simply reflect sample size variation, or there is(are)elements of the targeting cassette that are still left after Cre-mediated deletion that influence the expression of other genes in PWS or Angelman syndrome loci? For example, an influence on Necdin (Ndn) or Ube3a gene expression that is reported in this study. Although Ube3a was reported to be slightly affected (upregulated) in the PWScr model (PMID: 26848093).
RNA-seq analysis and its RT-qPCR verification raise reasonable doubts. Among verified genes only 50% of them showed tiny changes (if at all) with RT-qPCR between investigated genotypes.
Observed differences between PWS and PWS-KO mouse model during microbiome analysed should be explained in more details.
Minor
Supplementary table 1 is difficult to analyse. I would suggest to split it into sub-tables with proper table legend.
There are few typos like: PSW-KO (p9- 376), etc
It is unclear how the different phenotypes of mice were obtained. Were all of them, including wt, siblings or was each genotype, including wt, bred differently?
Reviewer 2 Report
This manuscript presents a wide-ranging analysis of the effects of conjugated linoleic acid (CLA) on the Snord116 deletion mouse model of Prader-Willi syndrome, a syndrome characterized by obesity and hyperphagia. The original rationale for the work stems from the observation that deletion of Snord116 reduces expression of Nhlh2, which itself regulates PCSK1, a prohormone convertase involved in maturation of several hypothalamic hormones. Dr. Good’s lab have previously generated a Nhlh2 knockout mouse that shares some phenotypes with PWS. Furthermore, CLA treatment of Nhlh2 deficient mice led to reductions in body fat, increased metabolism, and improved exercise performance, all of which would be beneficial to treatment of people with PWS. In this study, CLA was added to the diet of 9-week old wild type, mice bearing a Snord116 deletion on the paternal allele (Snord116 is maternally imprinted), or mice homozygous for the Snord116 deletion. Several measures were taken over the next 12 weeks and tissues then analyzed.
The major findings are that similar to WT, Snord116 deficient mice lose weight on CLA due to a decrease in fat mass. CLA increased fasting glucose levels of all genotypes, minimal effected respiratory exchange ratio, and potentially led to a small increase in energy expenditure. WT but not Snord116 deficient mice increased voluntary wheel running activity on CLA. Treatment increased muscle torque output for WT mice but a reduced torque output for paternal Snord116 deletion mice. Curiously, Snord116 deficient mice appear resistant to the hepatic steatosis caused by CLA treatment of WT mice. Results for a hypothalmic RNA-seq analysis treated and control mice WT and Snord116 deficient mice are presented. Lastly, cecal microbiome analysis reveals that Snord116 deficient mice have lower levels of Bacteriodetes, a correlate of obesity.
Overall, these experiments provide new information about the effects of CLA treatment and it potential as a treatment for PWS. The experiments were carefully performed and are cautiously interpreted. I have two comments for the authors:
1) Figure 2F shows food intake. Snord116 are smaller than WT littermates, presumably due to increased energy requirements to maintain body temperature below thermoneutrality. Are the presented results normalized to body weight and do the results change if this normalization is done?
2) There are now over 25 distinct engineered mutations at the mouse PWS locus (Ref. 39). As a reviewer, I support an effort to be more precise in publishing results on particular PWS alleles (Carias, 2019, Molecular Therapy: Methods & Clinical Development Vol. 13, https://doi.org/10.1016/j.omtm.2019.03.001. The use of “PWS” and “PWS-KO” is likely very confusing to readers not familiar the variety and distinctive traits of each model. I recommend the use of more descriptive and precise terminology for the genetic models used, such as Snord116 m+/p- and -/-.
Minor points:
Line 366. The S1E reference should be S1F
Line 376 This should be PWS not PSW
Line 562 The Figure 9 legend is truncated
Line 567 CL should be CLA
Round 2
Reviewer 1 Report
I have already suggested that this article be rejected because, in my opinion, it makes no scientific sense to investigate the effect of CLA on the growth of retired mice (the PWS mouse model), which show no signs of obesity. Also, I see no phenotypic similarity between N2KO and Snord116m+/- (or m-/-) mice. Importantly!!!, regarding Nhlh2 expression, the authors showed that there are no differences in hypothalamic Nhlh2 expression between WT and Snord116m+/-, yet they write the entire paper consistently suggesting Snord116 involvement in the regulation of Nhlh2 expression. Don't they really believe their own experimental data in this manuscript? All of the above led to my previous decision and has not changed in the revised version of the manuscript.
